# ThreadsGAN: Enhancing Coherence and Diversity in Discussion Thread Generation

## Abstract

Current research on generating discussion threads faces challenges in coherence, interactivity, and multi-topic handling, which are crucial for meaningful responses. This paper introduces ThreadsGAN, a model that enhances thread generation by incorporating multi-topic and social response intention tags. By leveraging BERT and Transformer, ThreadsGAN ensures contextual coherence and manages topic consistency. Additionally, it employs conditional generation to align responses with specific discussion contexts, and its CNN-based discriminator assesses response quality by evaluating similarity between generated and real responses, improving overall performance in generating realistic and contextually appropriate discussion threads.

## 1 Introduction

In contemporary society, social media and online forums have become dominant platforms for communication, with discussion threads emerging as the primary arenas for individuals to share their views and engage in discussions (Ahmed et al., 2019). However, as the volume of discussion threads grows rapidly, managing and maintaining this vast amount of information has become increasingly challenging. Many participants now face the problem of information overload. The advent of discussion thread generation models plays a crucial role in facilitating the dissemination and sharing of knowledge. These models help improve the learning efficiency of social media community members, enhance the linguistic expression within threads, and promote clearer and more persuasive communication. More importantly, they contribute to fostering constructive dialogues while mitigating the risk of meaningless or offensive remarks (Hamm et al., 2015). In this context, developing a model capable of generating discussion threads has become essential, as it effectively addresses the challenge of information overload and improves the quality and efficiency of discussions. Currently, research on discussion thread generation models has garnered significant attention in academia. Previous researchers have invested considerable effort in this field, attempting to develop models that can simulate real discussion scenarios and possess intelligent generation capabilities. Some of these studies have focused on natural language processing, deep learning, and generative adversarial networks, aiming to optimize the accuracy and diversity of thread generation. Consequently, research on question-answering dialogue generation models, which simulate human conversations, has flourished, laying a solid foundation for the further development of discussion thread generation models (Li et al., 2016).

However, the development of discussion thread generation models still faces a series of challenges, including whether the model can understand the context by simultaneously grasping the main post and subsequent responses, the authenticity and coherence of the generated outcomes, and the ability to adapt to diverse topics. Future research should concentrate on these challenges, enhancing the model's applicability across different scenarios by integrating more context-aware and sentiment analysis techniques (Aragón et al., 2017). In summary, research on discussion thread generation models holds significant practical value, impacting the improvement of thread quality, the enhancement of participant experience, and the evolution of social media and online forums. Through in-depth research and continuous innovation, the goal is to construct more intelligent discussion thread generation models that meet user needs. To address the challenges posed by offline discussion thread generation tasks, this study proposes a novel approach utilizing existing text generation models, referred to as ThreadsGAN. Specifically, this approach involves modifying the existing GAN model architecture, which will be detailed in the experimental methodology. Finally, to eval-

uate the effectiveness of this approach in addressing the problem, comparisons will be made with cutting-edge models, including Sequence Generative Adversarial Nets with Policy Gradient (Seq-GAN) (Yu et al., 2017) and Large Language Models (LLM) (Zhao et al., 2023) . Additionally, the study will thoroughly explain the data preprocessing methods and parameter tuning approaches, as the comparative models in this research employ different methodologies. To ensure the experiments proceed smoothly, these methods have been appropriately adjusted.

## 2 RELATED WORK

### 2.1 MASKED LANGUAGE MODELING

Easy data augmentation (EDA) utilizes techniques such as homophonic word replacement, synonymous word replacement, random insertion, and random deletion (Wei & Zou, 2019). While straightforward, these methods exhibit several limitations. Homophonic word replacement may unintentionally introduce inappropriate vocabulary, synonymous word replacement can subtly shift sentence semantics, and random insertion or deletion may result in unnatural sentence structures. More critically, in tasks such as generating coherent responses within discussion threads, EDA techniques are fundamentally inadequate. The reliance on simple word substitutions leads to responses that are only slightly varied and lack the contextual depth or logical progression needed to form meaningful dialogue sequences.

Masked language modeling, another EDA technique, attempts to address some of these issues by masking certain words in the text and replacing them with contextually appropriate alternatives. This approach offers improved semantic preservation and structural integrity compared to traditional EDA methods. However, despite its greater precision, masked language modeling still faces challenges in producing coherent and contextually consistent responses in sequential dialogue. As a result, both traditional EDA techniques and masked language modeling are limited in their ability to generate discussion threads that require a logical flow of interconnected responses.

### 2.2 GENERATIVE ADVERSARIAL NETWORKS (GAN)

Generative Adversarial Networks (GANs) have emerged as a powerful framework for generating synthetic data through a two-player game between a generator and a discriminator (Goodfellow et al., 2014). Due to their ability to capture complex data distributions, GANs have found widespread applications across various domains. However, it is crucial to acknowledge that different variants of GANs possess distinct strengths and limitations, contributing to their effectiveness and constraints in various applications. In exploring these variants, RelGAN prioritizes relationship modeling (Nie et al., 2018), aiming to enhance the network's understanding of intricate connections within the generated data. SentiGAN places a strong emphasis on emotion control (Wang & Wan, 2018), providing a valuable tool for applications requiring nuanced emotional expression in synthetic data. CatGAN introduces category information (Liu et al., 2020), a significant feature for tasks where categorization plays a pivotal role in data generation. Meanwhile, CycleGAN focuses on cross-domain transformations (Yuan et al., 2018), enabling seamless translation between diverse domains. Despite the remarkable performance of these models on specific tasks, it is essential to highlight their inherent drawbacks when compared to SeqGAN. RelGAN's main limitation lies in its insensitivity to the finer details of generated images, potentially impacting the fidelity of the synthetic data. SentiGAN, while effective, may encounter challenges related to achieving sufficient emotional diversity in its generated content. CatGAN's performance heavily relies on precise category information, thereby limiting its applicability to scenarios where such information is readily available. Although Cycle-GAN excels in image translation tasks, its direct applicability in generating sequential data, such as discussion threads, is somewhat constrained. In contrast, SeqGAN emerges as a more fitting choice for the task of generating discussion threads. Tailored specifically for handling sequential data, Seq-GAN excels in capturing the temporal nature of discussion threads and effectively managing context and coherence. This specialized design positions SeqGAN as an ideal solution for generating discussion threads, showcasing its capability to address the unique challenges posed by sequential data generation in conversational contexts.

## 2.3 LARGE LANGUAGE MODELS (LLMs)

Large Language Models (LLMs) have seen rapid advancements in recent years, demonstrating exceptional capabilities in various natural language processing tasks, such as text generation, translation, summarization, and conversational agents (Xi et al., 2023). Their ability to comprehend and generate human-like text makes them a valuable tool for a wide range of applications (Roumeliotis & Tselikas, 2023). Due to their proficiency in capturing contextual nuances and producing coherent, contextually relevant content, LLMs have become a benchmark in the field of text generation. In the context of generating discussion threads, incorporating LLMs as a comparative model is essential due to their superior performance in generating fluent and context-aware text. However, despite these strengths, LLMs are not without limitations. One significant drawback is their tendency to produce generic or overly verbose responses, which may lack the specificity and depth required for meaningful discussion threads. Additionally, LLMs often struggle with maintaining thematic consistency across extended conversations, leading to disjointed or irrelevant responses. Furthermore, LLMs can exhibit issues with factual accuracy, as they may generate hallucinations or incorrect information, particularly when dealing with niche or specialized topics. This can undermine the credibility of the generated discussion threads (Gao et al., 2024).

Another limitation of LLMs is their difficulty in accurately capturing the subtleties of user intent, especially in multi-turn dialogues where the context evolves dynamically. LLMs may fail to prioritize or emphasize key aspects of the discussion, resulting in a misalignment between user expectations and generated content. Moreover, while LLMs excel at generating human-like language, they are not inherently designed to adhere to the structural or topical requirements of specific domains, such as technical or expert-level discussions, which can result in responses that lack domain-specific relevance or rigor. Finally, the extensive computational resources required to train and deploy LLMs pose a significant barrier to their widespread adoption (Wan et al., 2023). Their performance heavily depends on pre-trained models, which are susceptible to biases embedded in the training data, potentially leading to biased outputs that skew discussions in unintended directions (Lin et al., 2024). These challenges underscore the importance of not solely relying on LLMs for discussion thread generation but rather using them as a benchmark to guide the development of more specialized, efficient, and contextually aware models that better align with the goals of targeted discussion thread generation.

## 3 PROPOSED METHOD

The Introduction presents a GAN-based functional architecture designed to generate discussion threads. This model includes a generator and discriminator, which compete to produce contextually relevant responses, forming a complete discussion thread when concatenated. Subsequent sections cover data collection, preprocessing, model architecture, generation process, evaluation metrics, and comparative methods. The architecture is further divided into detailed subsections on the generator, discriminator, and other key components.

### 3.1 DATA COLLECTION & PREPROCESSING

The dataset used in this study was collected from online community discussion platforms. Given the diversity and complexity of topics and content on these platforms, data collection focused on a limited number of specific categories. The selected data scraping targets were discussions that exhibited moderate disagreement and covered a range of topics. Ideally, the dataset includes contributions from highly engaged participants in the chosen discussions, ensuring data quality. Excessive disagreement, overly diverse topics, or irrelevant conversations could lead to generated content lacking coherence or meaningful discussion. After data collection, further filtering and cleaning were necessary to refine the data into a format suitable for training. Posts with fewer than 20 responses were excluded, as the likelihood of such responses exhibiting coherence, interactivity, and multi-topic aspects was deemed lower. Each post in the final dataset contains between 20 to 50 responses. This ensures that the selected posts meet the criteria essential for the model to learn key discussion thread characteristics, such as coherence, interactivity, and multi-topic handling. In addition to collecting the content of posts and responses, a "subject area" label was introduced. The subject area labeling was manually conducted by three experts using a multi-topic clustering method, where each response was individually labeled, with the final label determined by majority vote. How-

ever, in cases where consensus was not reached, a secondary review was conducted until agreement was achieved. Consequently, the final dataset consists of four components: post content, response content, previous response content, and the associated subject area.

## 3.2 MODELING

The proposed method advocates adjusting the existing GAN model architecture, with the Generator utilizing the BERT model and the Transformer Decoder as the primary base models. BERT is employed to extract hidden features from the original posts, followed by the Transformer Decoder, which generates responses based on the features extracted by BERT. Before generation, the normal distributions of the preceding and following responses are estimated separately, and their similarity is controlled to maintain coherence in the discussion content. Additionally, the hidden features of the main post are considered a primary condition. The Generator can generate probability distributions of responses, while the Discriminator first uses a CNN layer, followed by a MaxPooling layer, to analyze the similarity matrix between the generated response and the previous response, determining whether the relationship represents a genuine sequence or a fabricated next response. The following explanation will be divided into multiple paragraphs for a more detailed discussion.

### 3.2.1 INPUT

The data collected in this project is categorized into three types: the main posts of discussion threads, responses, and topic labels of responses. The details are as follows:

- `Post`: The format and content of the post input items in this year are consistent with those of the first year.

- `Response`: The format and content of the response input items in this year are the same as those used in the first year.

- `Topic`: The topic labels used in this year are consistent with the format and content from the second year.

Each training sample consists of four items: the post ($X^{POST}$), response ($X^{NEXT}$), previous response ($X^{PREV}$), and topic label ($X^{TOPIC}$). For example, when generating the first response, the previous response equals the post $x^{PREV}=X^{POST}$, $x^{NEXT}=x_1^{RESP}$, $x^{TOPIC}=x_1^{TOPIC}$, and for generating the second response, the previous response equals the first response $x^{PREV}=x_1^{RESP}$, $x^{NEXT}=x_2^{RESP}$, $x^{TOPIC}=x_2^{TOPIC}$.

### 3.2.2 GENERATOR

In the Generator, the BERT model is used to encode the main post and both preceding and following responses. The similarity of the preceding response is calculated within its probability distribution, serving as a control mechanism for maintaining topic coherence. Following this, the Transformer's decoder model is employed to generate responses, enabling the Generator to learn how to produce the next response in a discussion thread based on the content of the main post and surrounding responses.

**BERT hidden features** The main purpose of the BERT layer is to obtain the hidden features of each token in each response. This project will discuss inputting the main post ($X^{POST}$), the previous response ($X^{PREV}$), and the response following the current post ($X^{NEXT}$) into $BERT_{POST}$, $BERT_{PREV}$, and $BERT_{NEXT}$ models, respectively. After processing, the hidden features of the main post, the previous response, and the next response are obtained, denoted as

$$H^{POST} = BERT_{POST}(X^{POST}) \tag{3.2.1}$$

$$H^{PREV} = BERT_{PREV}(X^{PREV}) \tag{3.2.2}$$

$$H^{NEXT} = BERT_{NEXT}(X^{NEXT}) \tag{3.2.3}$$

Pre-trained BERT models were employed as the initial network weights for $BERT_{POST}$, $BERT_{PREV}$, and $BERT_{NEXT}$, effectively reducing overall training time and enhancing the model's prediction performance.

**Post, previous response, and next response feature representations.** Compared to the previously obtained response representation features, the standard BERT method was employed in this study to represent the features of the post, previous response, and next response, denoted as $h_{<CLS>}^{POST}, h_{<CLS>}^{PREV}, h_{<CLS>}^{NEXT}$.

**Prior and Recognition Estimation** In this study, the prior distribution of the previous response was assumed to be a multivariate Gaussian $p_\theta(z^{PREV}|h_{<CLS>}^{PREV}) = \mathcal{N}(\mu', \sigma'^2 I)$, where $I$ is a diagonal matrix, and $\mu'$ and $\sigma'^2$ represent the mean and variance, respectively. The mean and variance were estimated using an MLP layer, with the central equation as follows:

$$\left[\mu', \log(\sigma'^2)\right] = MLP_{PRIOR}(h_{<CLS>}^{PREV}) \tag{3.2.4}$$

where $MLP_{PRIOR}$ is a linear layer, and its output dimensionality matches that of $h_{<CLS>}^{POST}$. Similarly, the recognition posterior of the next response was assumed to be a multivariate Gaussian $p_\theta(z^{NEXT}|h_{<CLS>}^{NEXT}, h_{<CLS>}^{PREV}) = \mathcal{N}(\mu, \sigma^2 I)$, where $I$ is a diagonal matrix, and $\mu$ and $\sigma^2$ represent the mean and variance, respectively. The mean and variance were estimated using an MLP layer, with the central equation as follows:

$$\left[\mu, \log(\sigma^2)\right] = MLP_{RECO}(h_{<CLS>}^{NEXT}, h_{<CLS>}^{POST}) \tag{3.2.5}$$

where $MLP_{RECO}$ is a linear layer, and its output dimensionality matches that of $h_{<CLS>}^{POST}$.

**Decoder Generating the Next Response** In this project, the generation model for the next response ($DECODER_{DATAUG}$) was employed to simulate the structure of a discussion chain. The hidden representation of the generated response, $H'^{NEXT}$, is obtained using the following formulas:

$$H'^{NEXT} = MLP_{GENERATE}(H^{NEXT}) \tag{3.2.6}$$

$$H^{NEXT} = DECODER_{DATAUG}$$
$$(h_{<CLS>}^{POST} \oplus h_{<CLS>}^{PREV} \oplus z^{PREV} \oplus z^{NEXT} \oplus h^{TOPIC}) \tag{3.2.7}$$

$$h^{TOPIC} = MLP_{TOPIC}(x^{TOPIC}) \tag{3.2.8}$$

where $MLP_{TOPIC}$ is linear layer, and its output dimensionalities match that of $h_{<CLS>}^{POST}$. The dimension size of $H'^{NEXT}$ corresponds to the number of characters in the generated sentence. Each character of the generated response $x''^{NEXT}$ is derived from the following equation:

$$x''^{NEXT} = \arg\max(H'^{NEXT}) \tag{3.2.9}$$

### 3.2.3 DISCRIMINATOR

Based on the generated response probability vector $H''^{NEXT}$ obtained in the previous step, the goal of the Discriminator is to maximize $L_D$, ensuring it can distinguish between the real response $x^{NEXT}$ and the generated response $x''^{NEXT}$, making the generated response as distinct as possible from the real response.

Initially, the Discriminator converts the hidden vector $H''^{NEXT}$ from the generated response into the same dimensional space as the previous response $h_{<CLS>}^{PREV}$ using an MLP layer, producing $H^{\varphi NEXT}$. The hidden vectors $H^{\varphi NEXT}$ and $H^{PREV}$ are then multiplied to obtain the feature matrix $M_g$. This matrix is passed through CNN and MaxPooling layers for feature extraction, as expressed in the following equation:

$$H_{CNN} = MaxPooling(CNN(H^{\varphi NEXT} \cdot H^{PREV})) \tag{3.3.1}$$

$$H^{\varphi NEXT} = MLP_\varphi(H''^{NEXT}) \tag{3.3.2}$$

Finally, $M_g$ is passed through MLP and Sigmoid layers to obtain $m_g \in (0, 1)$. In addition, $H^{PREV}$ is combined with $H^{NEXT}$ to compute the similarity score $m_t$ using the same method.

$$m_g = MLP_\delta(H_{CNN}) \tag{3.3.3}$$

### 3.2.4 ACTUAL GENERATION OF DISCUSSION THREAD RESPONSES

At this stage, $BERT_{POST}$ was used to input the post, and $BERT_{PREV}$ was used to input the previous response. The hidden vector for the post, $H^{POST}$, and the hidden vector for the previous response, $h^{PREV}_{<CLS>}$, were obtained. Then, $h^{PREV}_{<CLS>}$ passed through $p(z^{PREV}|h^{PREV}_{<CLS>})$ to derive $z^{PREV}$. This, along with $h^{PREV}_{<CLS>}$ and the topic of the previous response $x^{TOPIC}$, was fed into the $DECODER_{DATAUG}$ to generate multiple responses with varying topics.

### 3.2.5 EVALUATION OF GENERATED DISCUSSION RESPONSES

In this study, the BLEU evaluation metric was used to assess the effectiveness of the generated responses. BLEU is a widely adopted automated evaluation metric for machine translation, which measures the overlap between the generated responses and the original responses, considering different n-gram measures. Additionally, the ROUGE-L metric was applied to further address the sequence order issues that BLEU does not account for. ROUGE-L evaluates the order of words and their co-occurrence in the generated responses compared to the original responses. Each generated response was evaluated using these metrics to assess the model's generation performance.

## 4 EXPERIMENTAL EVALUATION

### 4.1 DATA DESCRIPTION

The dataset utilized in this research has its origins in the parenting section of the Taiwanese community forum website, PTT (`https://term.ptt.cc/`). It was refined to ensure it contained high-quality discussions across 10 distinct subject areas, as illustrated in Figure 1. These subject areas were carefully selected to provide a diverse yet structured set of discussions, incorporating a range of perspectives while maintaining a level of coherence essential for model training. After applying filtering criteria, the final dataset comprised 920 posts and 29,754 responses. Figures 2 and 3 illustrate the distribution of word counts in posts and responses, respectively. With each post containing between 20 and 50 responses, Figure 4 illustrates the distribution of the number of comments per post. This distribution guarantees that the posts and responses retained for the model sufficiently exhibit the desired characteristics of coherence, interactivity, and multi-topic handling. The finalized dataset was then split into training and testing sets, with 23,803 responses in the training set and 5,951 responses in the testing set.

### 4.2 BASELINE MODEL

This section presents the evaluation of the proposed ThreadsGAN model in comparison with several baseline models that could be applied to the task of discussion thread generation. The baselines include sequence generation models and large language models (LLMs) such as TAIDE (`https://TAIDE.tw/`), an open-source model based on Taiwanese culture, and GPT-4 (Achiam et al., 2023), a commercial LLM. The models were evaluated using traditional metrics including BLEU (Papineni et al., 2002), ROUGE-1, ROUGE-L (Lin & Och, 2004), and BERTScore (Zhang et al., 2019), which assess different dimensions of the quality of generated threads.

The experimental results are summarized in Table 1, and the following key observations are made:

- `SeqGAN`: an early generative model for sequence generation, showed consistently low performance across all metrics. With ROUGE-1 and ROUGE-L scores of 1.67%, it demonstrated limited capacity to recall relevant content. The near-zero BLEU score and a

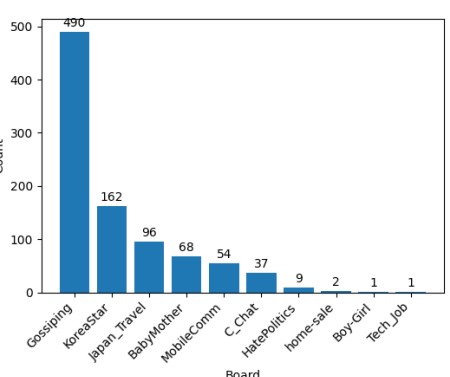

Figure 1: Board Value Counts with Count Displayed.

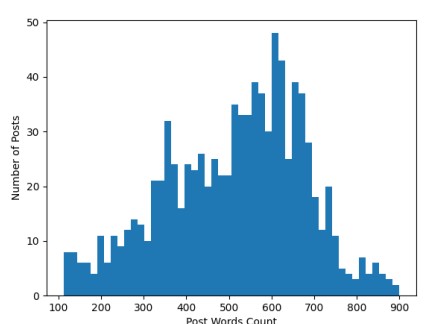

Figure 2: Distribution of Post Word Count.

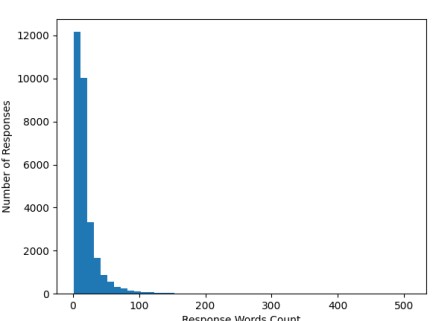

Figure 3: Distribution of Response Word Count.

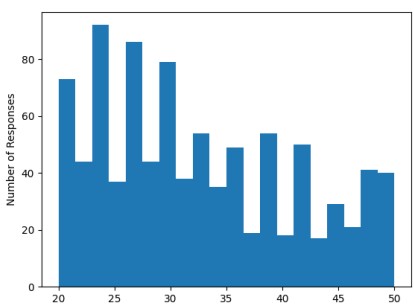

Figure 4: Distribution of Responses per Post.

BERTScore of 51.92% suggest that SeqGAN's ability to generate meaningful and coherent thread content is minimal, making it less suitable for complex thread generation tasks.

- `TAIDE`: despite being based on domain-specific (Taiwanese cultural) knowledge, exhibited only moderate improvement. It achieved ROUGE-1 and ROUGE-L scores of 7.13%, indicating a better recall of key information. However, its BLEU score was only 0.22%, reflecting difficulty in maintaining accurate n-gram sequences. The BERTScore of 31.33% highlights challenges in generating semantically coherent responses, suggesting that while TAIDE can produce culturally relevant content, its overall quality in generating cohesive and meaningful threads is limited.

- `GPT-4`: while representing a state-of-the-art LLM, displayed mixed results. It performed well in the BLEU metric (30.63%), reflecting its ability to produce content that matches human-written threads at the token level. However, its ROUGE scores (both 1.14%) were relatively low, suggesting that GPT-4 struggled with recalling detailed information across longer conversations. The BERTScore of 62.55% shows that while GPT-4 can maintain a degree of semantic relevance, it does not consistently generate content that aligns with human expectations in a discussion thread context, especially when deeper conversational flow and coherence are required.

- `ThreadsGAN`: the proposed method of this research, demonstrates a distinctive strength in generating semantically relevant and contextually coherent threads. Although the ROUGE and BLEU scores (ROUGE-1: 0.00%, BLEU: 0.12%) may appear modest, these metrics focus on token-level overlap and may not fully capture the essence of long-form, dynamic discussions that ThreadsGAN aims to generate. More importantly, ThreadsGAN achieved a BERTScore of 49.11%, suggesting that the model excels in generating responses that

maintain semantic consistency and align with the overall thread context, despite not prioritizing exact lexical overlap.

The results demonstrate ThreadsGAN's strength in generating semantically cohesive and contextually relevant threads, making it suitable for tasks like thread management, where conversational flow is prioritized. Despite facing challenges compared to large language models such as GPT-4, which benefit from vast datasets and extensive training, ThreadsGAN shows promise in producing coherent content within its domain.

Large language models excel in capturing complex linguistic patterns and generating fluent, semantically rich text due to their expansive data exposure. In contrast, GAN-based models like ThreadsGAN face limitations in semantic understanding and grammatical accuracy due to adversarial training. Nonetheless, ThreadsGAN proves effective for targeted applications that value content relevance and conversational coherence over lexical precision.

Table 1: Benchmark Results of Models

| Method | ROUGE-1 | ROUGE-L | BLEU | BERTScore |
|---|---|---|---|---|
| SeqGAN | 1.67% | 1.67% | 2.19e-80 | 51.92% |
| TAIDE | 7.13% | 7.13% | 0.22% | 31.33% |
| GPT-4 | 1.14% | 1.14% | 30.63% | 62.55% |
| ThreadsGAN | 0.00% | 0.12% | 0.12% | 49.11% |

## 4.3 HUMAN EVALUATION

Human evaluation is crucial for assessing the quality of generated discussion threads, as traditional metrics like BLEU and ROUGE fall short in capturing essential aspects such as coherence, interactivity, and multi-topic handling. Automated metrics often overlook the subtleties of natural conversation, making human judgment necessary for a comprehensive understanding of the models' effectiveness.

The evaluation focuses on five key criteria:

- `Coherence`: Ensures logical consistency within the thread, maintaining a natural flow between responses.
- `Interactivity`: Measures the response's ability to prompt further discussion and engagement.
- `Diversity`: Assesses variation in responses, preventing repetitive or formulaic content.
- `Readability`: Ensures linguistic clarity and grammatical correctness for easy comprehension.
- `Relevance`: Confirms that the responses are aligned with the discussion topic and context.

The evaluation process involved randomly selecting 30 generated responses, which were then rated by an expert using a 1-5 scale, where 5 represents the highest performance. This approach provided valuable insights that automated metrics could not, offering a more detailed and context-aware assessment of the generation methods.

Table 2: Human Evaluation Results of Models

| Method | Coherence | Interactivity | Diversity | Readability | Relevance |
|---|---|---|---|---|---|
| SeqGAN | 4.50 | 3.50 | 1.10 | 1.25 | X |
| TAIDE | 4.70 | 3.25 | 4.50 | 1.10 | 3.50 |
| GPT-4 | 4.70 | 5.00 | 5.00 | 4.50 | 5.00 |
| ThreadsGAN | 3.50 | 3.50 | 5.00 | 1.25 | X |

The human evaluation results (refer to Table 2) indicate that ThreadsGAN excels in generating diverse responses, achieving an impressive diversity score of 5.00. This high diversity is primarily

due to the model's multilingual output, frequently combining Chinese, English, and Japanese. However, this also accounts for its low readability score of 1.25, as the mixture of languages complicates comprehension.

Despite its low readability, ThreadsGAN maintains a moderate coherence score of 3.50, suggesting a reasonable level of logical flow in its responses. Nonetheless, its outputs were marked as irrelevant (denoted as "X" in relevance), likely due to the challenges in meeting human expectations and the complexity of its multilingual nature.

In contrast, GPT-4 achieves top scores in coherence, interactivity, and relevance, demonstrating its superior ability to produce fluent, contextually appropriate content. While ThreadsGAN excels in diversity, its struggles with readability and relevance suggest it is more suitable for tasks where content variety is prioritized over grammatical clarity.

## 5 CONCLUSION

This study offers significant contributions through the development of ThreadsGAN, a model designed to generate semantically coherent and contextually relevant discussion threads. The model is particularly effective for managing tasks where maintaining the logical flow of a discussion takes precedence over exact lexical matching. However, human evaluations reveal limitations, particularly in readability and relevance, due to the complexity of its multilingual outputs, which hinders overall comprehension.

These limitations point to the need for improving the model's handling of multilingual content and better aligning its outputs with user expectations in terms of relevance. Furthermore, traditional evaluation metrics such as ROUGE and BLEU are inadequate for assessing the coherence of discussion threads, emphasizing the necessity for more advanced, semantic-based metrics. Future research should focus on refining ThreadsGAN's ability to manage long-term coherence within discussions, enhance relevance, and develop more suitable evaluation frameworks for thread generation tasks.

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
