# OpenReview forum: "ThreadsGAN: Enhancing Coherence and Diversity in Discussion Thread Generation"
_ICLR.cc/2025/Conference — Submitted to ICLR 2025_

### Official Review · Reviewer_yrYX · 2024-10-28

**Soundness:** 1
**Presentation:** 1
**Contribution:** 2
**Rating:** 3
**Confidence:** 4

**Summary:**

The paper proposes ThreadsGAN, a model built with BERT, transformer, CNN and GAN, aiming for incorporating multi-topic and social response intention tags to tackle thread generation task. The paper claims that the proposed model has strong performance for generating realistic and appropriate discussion threads.

**Strengths:**

1. ThreadsGAN integrates GAN with topic and response tags to enhance discussion thread coherence and diversity. It is novel to use GAN as a text generation model.
2. The paper collected dataset from PTT website, which is across multiple subjects and has a large size of threads. This data is useful to train and evaluate models on thread generation task.

**Weaknesses:**

Overall this paper needs a major revision. I list my comments below:
1. Weak motivation. I am not convinced by the paper why thread generation is an important task to tackle with. The paper did not cite any related work in thread generation (Line 35), nor explained the usability or necessity for downstream application. How thread generation is different from conversation generation, or question answering? Why previous works cannot solve this task?
2. Lacks clarity in methodology. The paper did not formally explain the task - what is expected input and output. The authors explained the model's input in Section 3.2.1, but without a formal definition of the task it was hard to understand. For example, the paper mentioned 'year' the first time in 3.2.1, which wasn't mentioned before, and the role of year is not explained.
3. Missing explanation of model design. The proposed model has a lot of components, like BERT, CNN and GAN. There are a lot of alternatives that are generally better than BERT (RoBERTA) and CNN (LSTM), but why using these models as modules are not explained nor compared.
4. Weak performance: Both automatic and human evaluation showed that the proposed model is worse than GPT4, and even SeqGAN in some criteria. It is not explained why the proposed method better (if not in performance, then in cost, explanability...), or takeaways of using GAN as a generation model. Furthermore, the baselines are too weak to compare. The paper used unprompted, off-the-shelf GPT-4 (since the paper did not mention any adjustment), which I think will perform stronger with a well-designed prompting or SFT.

**Questions:**

1. What is the key benefit from using GAN as a text generator, compared with SoTA LLM?

---

### Official Review · Reviewer_xFxe · 2024-11-02

**Soundness:** 1
**Presentation:** 2
**Contribution:** 1
**Rating:** 3
**Confidence:** 4

**Summary:**

This paper proposes a GAN-based method to train a generative model that generates discussion threads. The model architecture is largely based on BERT but modifications were made to take into consideration the thread structure, explicitly separating the main post, the previous response, and the next response. A hidden representation of topics is also employed into the generation process. A discriminator model is trained concurrently to distinguish generated threads and real threads to help improve the generation model. Evaluation is done on data collected from a Taiwanese online forum. The proposed method is compared with SeqGAN and LLMs. The results are mixed. The proposed method doesn’t outperform GPT-4 but performs better than some other baselines based on some of the metrics.

**Strengths:**

The paper proposes a new customized method for a special task.

**Weaknesses:**

- The proposed method is not sound, based on my understanding.

Although the high-level idea of using GAN to develop a thread generation model is fine, the model details seem to have some issues. Specifically, the encoder takes in the next response as input (formula 3.2.3), but the decode generates the next response (formula 3.2.9), that is, the output is given as the input. Isn’t this a problem? I might have misunderstood the model, but the description of the model is not clear enough.

- The presentation needs improvement.

As mentioned above, the description of the method is not clear enough to resolve the main question I have about the validity of the generation process. Other than the method section, the other sections also lack important details and are sometimes not clear. For example, how are LLMs used to generate the threads? Do you use a prompt and if so, how does the prompt look like?
Most importantly, while the introduction section discusses several challenges for discussion thread generation, such as ensuring authenticity and coherence of the generated posts and ability to adapt to diverse topics, it is not clearly explained why the proposed method addresses these challenges.

- Lack of motivation.

The task studied is not well motivated. The proposed solution is not linked to the identified challenges and therefore is not well motivated, either.

- The work has limited contribution.

First, it is not clear to me why discussion thread generation is important and what its applications are. Although the authors try to motivate the work in the introduction section, there are very limited references supporting the claims that discussion thread generation is important and that much work has been done on this problem. The few cited papers are also old and do not represent the state-of-the-art solutions to text generation problems. The proposed method itself has some issues as pointed out above in the soundness section. In addition, the results do not show advantage of the proposed method over LLMs, which are now widely adopted for many tasks. Therefore, it is hard to imagine what other researchers can learn from this work.

**Questions:**

- What are some examples of applications of discussion thread generation?
- I am not sure whether there has been a large body of work on discussion thread generation in recent years. Could you provide references to recent papers that demonstrate the importance of discussion thread generation as stated in the first paragraph of Section 1?
- How are LLMs used to generate threads?

---

### Official Review · Reviewer_3zKn · 2024-11-03

**Soundness:** 1
**Presentation:** 1
**Contribution:** 1
**Rating:** 1
**Confidence:** 5

**Summary:**

The paper presents ThreadsGAN, a generative model designed to improve the quality of discussion threads by focusing on coherence, interactivity, and diversity. The model is evaluated against several baseline models, including SeqGAN, TAIDE, and GPT-4, using both traditional metrics and human evaluations.

**Strengths:**

N/A

**Weaknesses:**

The quality of the paper is very poor. The logical coherence between the paragraphs of the paper is poor and does not conform to the conventional writing style of a paper. The method proposed in the paper lacks any innovation. The experimental data in the paper also shows that the proposed method has poor performance.

**Questions:**

N/A

---

### Official Review · Reviewer_U518 · 2024-11-05

**Soundness:** 2
**Presentation:** 1
**Contribution:** 1
**Rating:** 1
**Confidence:** 4

**Summary:**

This submission examines the generation of discourse discussions threads using a Generative Adversarial Network (GAN).  The authors condition on topic and social response tags to help guide the generation process.  The work grounds on a Sequence GAN model for discourse and applies it to a parenting forum found in Taiwanese social media, and validate using human evaluation.

**Strengths:**

* The work examines an interesting application of discussion thread generation.
* The model uses relatively lightweight models for the characterisation (BERT) and generation, incorporating a CNN layer.
* The work performs a human evaluation of the model and baselines.

**Weaknesses:**

* The related work section needs more contextualisation before getting into the specifics of individual works.  It is not clear of the organisational structure and overall relevances of which works were reviewed and why. How they are organised and presented is crucially missing -- e.g., the GAN (S2.2) section is a mismatch of single sentence summaries that do not present a coherent argument for understanding the prior work and how it motivates the current submission.
* The provenance of the data (source platform and actual chosen topics) was not clear from (L140-161).  The quality guidelines and how interannotator agreement on the application of the guidelines was not elaborated on, so the veracity and quality of the data are suspect.
* There are design choices that are not well defended.  The authors decide to use BERT as part of their architecture, but choose to use a decoder-only transfomer architecture for generation, and the more natural choice of using a compatible embedding model that would be congruent with the decoder backend, was not defended.  I would need a more strong motivation and defense of the design choice.
  * Similarly, the use of a CNN for thread generation is not defended.  The work is generating a sequence of tokens, so it's not clear why a patch oriented model (such as a CNN) is used.
* The dataset is somewhat limited and exactly why the authors applied the work only to this particular domain and discussion board was not clear.  Fig. 1 further shows 4 of the 10 boards as having fewer than 10 viable posts, so it's not clear that the study's output generalises well.
* The descriptions of the baselines are insufficient to allow reproducibility.
* The ThreadGAN model does not perform well against commercial methods (GPT-4) or its basic baseline (SeqGAN), which would need to be demonstrated for publication.
* The human evaluation does not discuss inter-annotator agreement, which is a concern.  Without this aspect the quality of the evaluation is not assured, and the veracity and seriousness of the human evaluation is undermined.

**Questions:**

Generally, I feel this paper is not yet ready -- the front and back halves of the paper do not come together to support each other well.  The paper would do with more editing and refinement.  For example the conclusions mention shortcomings with multilingual output but these issues are not present at all at the outset of the work.
For these reasons, I am recommending rejection.

* There is prior work on discourse thread representation that may be applicable to your work in characterising/generating hierarchies, even before LLMs:

    * Muthu Kumar Chandrasekaran, Carrie Demmans Epp, Min-Yen Kan and Diane Litman (2017). Using Discourse Signals for Robust Instructor Intervention Prediction. In Proceedings of the Thirty-First AAAI
    * Kishaloy Halder, Min-Yen Kan and Kazunari Sugiyama (2019) Predicting Helpful Posts in Open-Ended Discussion Forums: A Neural Architecture, In Proc. of the North American Chapter Meeting of the Association for Computational Linguistics (NAACL '19). Minneapolis, USA: June 2-7, 2019.

* (More of a concern) The motivation for a GAN to create discussion threads isn't well motivated.  There could be use cases in academic study that would be better as a frame for the work.  The work (L037) tries to motivate it by referring to past works, but without citation.  Currently the work is structured just as an academic exercise "Can we use GANs to create discourse threads?" but this is at best, weak.  Better to connect the atmospheric first paragraph of the work against the real outcomes of the work later.
* The GAN math is a somewhat straightforward application but exactly how the authors' GAN is different from the standard SeqGAN needs to be clearly outlined.  Where math is presented, it should make a difference in the discussion -- which I currently feel it does not.

Specific line-by-line comments:
* 058: extra space at the end of the sentence.
* 067: it's not clear why these works on EDA are included.  How are they related to thread generation at large?  We need the context before the details.
* 103: SeqGAN emerges in the middle of this paragraph as the motivated underlying backbone model, but this should be written in a more clear style.
* 160: use LaTeX quotation (it seems you are using LaTeX to compose your paper).
* 209-213: I didn't find these equations a particularly good use of your space.  Perhaps mention in text?

---

### Meta-Review · Area_Chair_weMP · 2024-12-19

**Metareview:**

The paper proposes ThreadsGAN, a model built with BERT, transformer, CNN and GAN, aiming to incorporate multi-topic and social response intention tags to tackle thread generation tasks. The paper claims that the proposed model has a strong performance in generating realistic and appropriate discussion threads. The focus is on improving coherence, interactivity, and diversity in generated threads. ThreadsGAN employs a discriminator model to distinguish between real and generated threads consisting of the main post, previous response, and next response, thereby improving the generator's performance.

**Strengths identified**
1. The paper to some extent tackles an interesting and unique task: generating discussion threads.

2. The integration of GANs with topic and response tags for generating discussion threads is interesting.

3. The dataset collected from the PTT website is diverse, covering multiple subjects and containing a large volume of threads, making it valuable for training and evaluation.

**Weaknesses that need to be addressed**
1. Lack of a clear justification for the importance of the thread generation task (in contrast to conversation generation).

2. Model design choices are poorly defended

3. Much improvement is needed in terms of clarity and presentation.

4. Lack of discussion about why the model underperforms on some of the aspects (as compared to baselines) or what unique benefits it offers, such as lower cost or improved interpretability.

**Additional Comments On Reviewer Discussion:**

Authors did not provide any rebuttal response.

---

### Decision · Program_Chairs · 2025-01-22

Reject